# Screening of Multitarget Compounds against Acetaminophen Hepatic Toxicity Using In Silico, In Vitro, and In Vivo Approaches

**DOI:** 10.3390/molecules29020428

**Published:** 2024-01-16

**Authors:** Muhammad Ali, Esha Asghar, Waqas Ali, Ghulam Mustafa, Irfan Aamer Ansari, Saadiya Zia, Siddique Akber Ansari, Sumaiya Khan

**Affiliations:** 1Department of Biochemistry, Faculty of Sciences, University of Agriculture Faisalabad (UAF), Faisalabad 38040, Pakistan; aliwaqas2237@gmail.com (W.A.); dr.saadiyazia@uaf.edu.pk (S.Z.); 2Department of Biotechnology, Akhuwat Faisalabad Institute of Research Science and Technology (A-FIRST), Faisalabad 38000, Pakistan; eshaasgher@gmail.com; 3Department of Biochemistry, Government College University Faisalabad (GCUF), Faisalabad 38000, Pakistan; drghulammustafa@gcuf.edu.pk; 4Department of Drug Science and Technology, University of Turin, 10124 Turin, Italy; irfanaamer.ansari@unito.it; 5Department of Pharmaceutical Chemistry, College of Pharmacy, King Saud University, Riyadh 11451, Saudi Arabia; sansari@ksu.edu.sa; 6Department of Drug Chemistry and Technologies, University “La Sapienza”, 00185 Rome, Italy; sumaiykhan10@gmail.com

**Keywords:** multitarget, quercetin, curcumin, acetaminophen, *Curcuma longa*, *Cinnamon zeylanicum*, antioxidant activity, molecular docking

## Abstract

Combination therapy and multitarget drugs have recently attracted much attention as promising tools to fight against many challenging diseases and, thus, represent a new research focus area. The aim of the current project was to screen multitarget compounds and to study their individual and combined effects on acetaminophen-induced liver injury. In this study, 2 of the best hepatoprotective multitargeting compounds were selected from a pool of 40 major compounds present in *Curcuma longa* and *Cinnamomum zeylanicum* by using molecular docking, ADMET profiling, and Pfizer’s rule of five. The two selected compounds, quercetin and curcumin, showed a high binding affinity for the CYP2E1 enzyme, MAPK, and TLR4 receptors that contribute to liver injury. The candidates caused the decreased viability of cancer cell lines (HepG2 and Huh7) but showed no effect on a normal cell line (Vero). Examination of biochemical parameters (ALT, AST, ALP, and bilirubin) showed the hepatoprotective effect of the candidate drugs in comparison with the control group, which was confirmed by histological findings. Taken together, quercetin and curcumin not only satisfied the drug-like assessment criterion and proved to be multitargeting by preventing liver damage but also showed anticancer activities.

## 1. Introduction

In today’s modern era, we are dependent on prescribed drugs, which can result in severe diseases. Drug-induced liver injury (DILI) is an uncommon, but potentially fatal, cause of liver disease that is associated with suggested medications, non-prescription drugs, and dietary supplements. According to published numbers, DILI is estimated to have an annual incidence of 10 to 15 per 10,000 to 100,000 persons exposed to prescription medications. About 44,000 people in the United States will experience DILI annually [1]. Medications account for >50% of these, with 37% of cases attributable to APAP [2]. Thus acetaminophen (APAP) is a major cause of abrupt liver failure.

Owing to the adverse effects of prescribed drugs [3], natural herbal remedies are gaining more popularity in treating various diseases as they have the ability to prevent and minimize the danger of a variety of types of oxidative damage with few or no complications. The plant which yields the maximum amount of phytochemicals could be a good contender for liver injury treatment [4].

Acetaminophen is metabolized by conjugation with sulfate and glucuronate, which are inert and excreted in the urine. However, when the activity of normal metabolic pathways is suppressed, excess APAP is metabolized by cytochrome P450 (CYP) proteins (particularly CYP2E1) and forms the highly reactive & highly toxic intermediate N-acetyl benzoquinone imine (NAPQI), which produces oxygen species (ROS) [5,6]. Neutralization and metabolism of NAPQI to the safer mercapturic acid is attained with the instant release of glutathione. In any case, once GSH is exhausted, the residual NAPQI causes hepatocyte damage due to an increased content of mitochondrial protein adducts, and the consequent oxidative stress caused by peroxynitrite, lipid peroxidation, and superoxide radicals [7,8]. In addition, an inflammatory response leads to programmed injury by the activation of c-Jun N-terminal kinase (JNK) and mitogen-activated protein kinase (MAPK) [9,10]. It has been proven that in APAP-induced liver injury, hepatic injury occurs when both the inflammatory response [9] and oxidative stress [7] are involved.

*Curcuma longa* and *Cinnamomum zeylanicum*, commonly known as turmeric and cinnamon, are widely used as both herbs and spices [11,12]. Phytochemical constituents isolated from *C. longa* and *C. zeylanicum* include curcumin, dimethoxy curcumin, bisdemethoxycurcumin, curcumenol, ar-turmerone, vanillic acid, beta-sesquiphellandrene, 1-(4-hydroxy-3,5-dimethoxyphenyl), cinnamate, quercetin, trans-2-hydroxycinnamic acid, sinapic acid, gallic acid, cinnamic acid, and cinnamyl acetate [13,14]. The latest pharmacological studies suggest that cinnamon and turmeric possess antidiabetic, antimicrobial, anticancer, and anti-inflammatory properties due to the presence of natural bioactive compounds, making them valuable and adaptable plants with a wide range of therapeutic qualities [12,15].

Numerous research studies have examined curcumin’s bioactivity and health benefits, including its effects on the immune system, cancer prevention, hepatoprotection, neuroprotection, cardiovascular protection, and antioxidative and anti-inflammatory properties [16]. Quercetin is recognized as a chemical with anti-inflammatory, anti-obesity, cardioprotective, and antioxidative properties. It is believed to be helpful in preventing neurological disorders, cancer, diabetes, heart disease, obesity, allergic asthma, and atopic illnesses [17].

It is highly challenging to explain the molecular basis and mechanism of action of medicinal and edible plants using conventional pharmacological approaches due to the presence of numerous bioactive chemicals that can exert pharmacological effects through a variety of routes and targets [18]. With the development of bioinformatics and molecular docking tools, it has become easier to screen for new drugs. Many reported receptors are significant in generating inflammatory responses and, thus, are potential targets for drug-induced liver injury. Molecular docking has been employed for more than three decades, resulting in the discovery and development of numerous drugs [19]. The first stage in drug research and development is usually to find active compounds from existing substances. Several pharmaceutical corporations have libraries with hundreds of chemicals, but maintaining the library and executing high-throughput screening are both costly. Virtual screening provides a faster way to test millions of chemicals in several days. Molecular docking is one of the most-often utilized virtual screening procedures when the target protein’s 3D structure is available and used as a receptor [20].

Recent research has focused on finding polypharmacological drugs that treat complicated (multifactorial) illnesses like cancer, neurological disease, and specific infections by acting on multiple targets. Molecular docking is a computer-aided drug design technique [21] that is one of the computational methods employed in the hunt for multifunctional medications.

The use of bioactive compounds to reform medicines for curing multiple diseases in the future have shown encouraging results [22]. In the current study, several antioxidative tests were performed, and the compounds were screened through an in silico analysis involving molecular docking. The compounds were docked with liver-injury protein receptors as mentioned above. The chosen multitarget compounds from the in silico study were then evaluated by an anticancer analysis, with a final examination in a liver-injury mouse model. In this model, we investigated the individual and combined degrees of protection of pure constituents against the hepatotoxicity of acetaminophen in mice. AST (aspartate aminotransferase), ALT (alanine aminotransferase), ALP (alkaline phosphatase) as well as bilirubin (markers of liver function) were tested in the serum of mice [23]. To check the hepatoprotective effect, histopathological analysis was also carried out.

## 2. Results

### 2.1. Percentage Yield

To obtain the highest yield from the plant and to enhance the antioxidant level in our extract, the extraction was performed with 80% ethanol [24]. The % yield from the rhizomes of *C. longa* and bark of *C. zeylanicum* were determined and listed in Table 1. Statistical analysis showed that the % yield from the rhizomes of *C. longa* was higher than that from the bark of *C. zeylanicum*.

### 2.2. Total Phenolic Content (TPC)

Phenolics are essential plant components responsible for antioxidative activity. In plant extracts, their hydroxyl groups facilitate the free-radical scavenging activity [25]. The TPC determined in *C. longa* and *C. zeylanicum* was expressed in unit milligram of Gallic acid equivalents per gram of each extract, and the results are shown in Table 2 and Figure 1.

### 2.3. Total Flavonoid Content (TFC)

Like phenolics, flavonoids are secondary metabolites of plants that are well known for their antioxidative activity [25]. The content of flavonoids observed in *C. zeylanicum* and *C. longa* was expressed in unit milligram of gallic acid equivalents per gram of each extract, as shown in Table 2 and Figure 1. Analysis showed that a higher concentration of flavonoids is present in *C. longa* than in *C. zeylanicum.*

### 2.4. DPPH Radical Scavenging Activity

The most-often-used method to determine the antioxidant potential is to measure DPPH radical-scavenging activity. The DPPH radical-scavenging activity indicates the presence of phenolics and flavonoids [26]. In this test, the extract of *C. zeylanicum* showed higher activity compared with *C. longa* alone or in combination. The effect of the plant extracts on the DPPH scavenging activity is shown in Table 3 and Figure 2.

### 2.5. Reducing Power Assay

The principle of the reducing power assay is that substances with reduction potential will react with potassium ferricyanide to form potassium ferrocyanide ions, which then react with ferric chloride to form ferric–ferrous complexes. The reducing power of *C. longa* and *C. zeylanicum* was 2.41 ± 0.17and 1.88 ± 0.05 milligrams per gram of plant extract, respectively, using ascorbic acid as a standard. Individual extracts showed activity, as did the combined extracts (Table 4 and Figure 2).

### 2.6. Cell Viability Assay 

In end-stage liver damage, the cells start multiplying in an uncontrolled manner; if severe, this leads to hepatocellular carcinoma, simply known as liver cancer [27]. The anticancer activity of quercetin and curcumin was assessed using the cell viability assay, indicating that their individual and combined application had significant cytotoxic activity against the Huh-7 and HepG2 cell lines while no effect on a normal cell line was observed. Quercetin and curcumin showed a significant decrease in the viability of cells, as observed for the standard anticancer drug doxorubicin as shown in Figure 3.

### 2.7. Molecular Docking 

#### 2.7.1. Screening

Using the molecular operating environment MOE15 [28], 20 compounds were screened from a library of 40 compounds on the basis of the S-score, as shown in the Appendix A. Each ligand showed varied binding scores and amino acid interactions with receptors of liver injury. MOE was only used for screening purposes as AutoDock [29] was better at showing and explaining the interactions compared with MOE. 

#### 2.7.2. ADME and Drug-Likeness Analysis

SwissADME is an online program that is used to compute molecular characteristics. It predicts compounds with diverse parameters like solubility and molecular weight. Additionally, Lipinski’s rule of five was applied. Lipinski’s rule of five states that a drug molecule generally does not violate more than one of the following five rules: molecular mass < 500 Da, high lipophilicity (expressed as LogP < than 5), <5 hydrogen-bond donors, <10 hydrogen-bond acceptors, and molar refractivity between 40 and 130. As displayed in Table 5, curcumin and quercetin did not violate any rules, suggesting that they could be suitable for oral administration. This indicates that curcumin and quercetin are drug-like candidates. Four compounds were selected by investigating the drug metabolism through enzymatic metabolism in the liver; our candidate drugs showed no violations while Rutin showed three violations. Finally, by fulfilling the ADMET profiling criteria, these were predicted as having probable drug-like characteristics, as shown in Appendix A. The compounds that were selected to proceed with further have good affinity and efficacy for receptors of liver-injury pathways. Despite all their advantages, the computational biology approaches have certain drawbacks, as several tools give different results for the same analyses; therefore, one cannot fully depend on the results without a wet-lab investigation and validation.

#### 2.7.3. Docking Analysis

Molecular docking, used in drug discovery, enables the identification of novel therapeutic compounds as it uses a multitargeting approach with disease receptors [30]. As visible within the molecular surface site of the protein unit, these three compounds—curcumin, quercetin, and silymarin—showed interactions with the residues of CYP2E1, MAPK, and TLR4. Overall, the binding mode of curcumin and quercetin with the residues showed stabilization through various interactions. Proline was most prominent residue to show pi–alkyl bonding with ligands, while glutamate and glycine commonly showed hydrogen bonds. Apart from pi–alkyl and hydrogen bonds, pi–pi stacked, pi–sigma and pi–anion bonds were also observed. The large number of hydrogen bonds and other interactions indicates the strong binding affinity of curcumin and quercetin, as shown in Figure 4, Figure 5 and Figure 6.

### 2.8. Liver to Body Weight 

During the APAP administration period, no animals died. In comparison with the control, a noteworthy decrease in the body mass and liver weight of APAP-administered mice was observed. The selected pure compounds or their combination were given to mice for fourteen days and no significant changes observed in mice weight (Figure 7).

### 2.9. Assessment of Liver Function

#### 2.9.1. ALT, AST, ALP, and Bilirubin

The extent of liver injury was assessed by biochemical markers of the liver (ALT, AST, ALP, and bilirubin). The levels of these markers increase in the case of liver injury [31]. In the APAP-treated group, increased values of these markers, except bilirubin, indicated severe hepatic tissue damage. Our pretreatment groups showed a significant decrease in the levels of these markers and showed comparable results to silymarin, as shown in Figure 8.

#### 2.9.2. Analysis of TAC and TOS

Oxidative stress contributes to the commencement and progression of liver injury; thus, to evaluate damage, several oxidative markers have been developed [32]. Biochemical parameters TAC and TOS from serum samples and the statistical evaluation between the groups are shown in Figure 9. There was a decrease in TAC levels as well as an increase in TOS observed in the APAP group. In the serum samples of treated mice, the level of TAC increased significantly compared with the APAP group. The TOS level decreased in treatment groups, and they were also statistically significant [33,34].

#### 2.9.3. Histopathology

Histopathological analysis showed a clear effect of the different treatments on liver tissue. The APAP-treated group showed severe hepatocellular necrosis, inflammation, and disrupted architecture, consistent with APAP-induced hepatotoxicity. Quercetin- and curcumin-pretreated groups showed milder histological changes compared with APAP, suggesting a potential hepatoprotective effect. The group receiving a combination of both compounds showed moderate necrosis, inflammation, and disrupted architecture, suggesting an additive or synergistic effect of the combined treatment. Small changes were observed in the silymarin group, suggesting a possible protective effect against liver damage. These findings suggest that quercetin, curcumin, and silymarin may have hepatoprotective properties, with silymarin having the mildest effect. The results are shown in Figure 10.

## 3. Discussion

The current project was designed to screen for multitarget hepatoprotective pure compounds from *C. longa* and *C. zeylanicum* [15] and to check their activity using in vitro and in vivo models for liver injury. Other linked assays like antioxidative and cell viability assays were also performed to demonstrate their potential as good therapeutic agents. 

The liver is the main site of metabolism for most xenobiotics, so the generation of free radicals in the liver is higher than in other organs [35]. ROS generated during the metabolism process of xenobiotics [36] cause hepatic cell injury. Thus, antioxidants play a role in protecting from injury by scavenging free radicals. Phenolics were found to be high in *C. zeylanicum* (117.5 ± 0.39 mg of Gallic Acid E/g of Extract), and a high content of flavonoids was found in *C. longa* (98.37 ± 0.27 mg QE/g of Extract). The percentage inhibition of DPPH and the reducing power of individual and combined extracts were determined but there was no synergistic activity seen. *C. zeylanicum* showed higher antioxidative activity (45.16 ± 0.66%). The results demonstrated that a high content of phenolics contributed more to the antioxidative activity than did flavonoids. The results were confirmed by previously available data [37,38,39,40,41].

The major goals in finding a new drug are potency and protection, as all drugs can help to treat diseases as well as produce adverse effects. The basic screening of compounds was performed by MOE analysis of a library of 40 compounds based on their S-score. Only two compounds (i.e., curcumin and quercetin) were shortlisted from the MOE analysis, which was followed by Lipinski’s rule of five (Ro5) and an analysis of associated ADMET properties by the admetSAR and SwissADME online tool. The associated ADMET properties of potential compounds for different models, such as P-glycoprotein substrates, BBB penetration, gastrointestinal absorption, metabolism (including of cytochrome inhibitors), mutagenicity, and carcinogenicity showed positive results that strongly support the compounds’ suitability as drug candidates. A drug for therapeutic purposes can be selected by considering its metabolism. Cytochrome p450 is very important in drug metabolism. Various cytochromes are present, but CYP1A2, CYP2C9, CYP2D6, CYP3A4 and CYP2C19 are the important ones in drug metabolism. The clearance of compounds by these parameters indicate the safety of the compound. In our current findings, quercetin and curcumin are safe and tolerable. Our finding are supported by reported pharmacokinetics analyses of quercetin [42] and curcumin [43] compounds. The computational biology approaches have certain drawbacks and, therefore, one cannot fully depend on the results without the addition of wet-lab investigation and validation.

As reported in the literature, the antioxidative activity of plants is due to the presence of multiple bioactive compounds. Docking analysis specifically selected the multitargeting compounds [13,14]. Binding to more than one of the receptors/enzymes CYP2E1, MAPK, and TLR4 proved the multitargeting ability of the compounds. Our compounds specifically showed high interaction with cytochrome P450 enzymes (CYP2E1 and CYP2C9). Cytochromes are mainly involved in the metabolism of exogenous and endogenous compounds and drugs, especially for the metabolism of acetaminophen. Hepatic CYP2E1 plays a major role in APAP-induced liver injury [44]. Compared with wild-type mice given acetaminophen, CYP2E1-knockout mice had a significantly higher resistance to liver damage [45]. Weber et al. reported that TLR4 deficiency protects against hepatic injury in preclinical mice models [46]. TLR4 is directly involved in hepatic inflammation and fibrosis, whereas upregulation of inflammatory factors like NF-κB by TLR4 is also important in hepatocarcinogenesis [47]. Our docking analysis showing that the candidate drugs inhibit the TLR4 cell-surface receptor, because its activation causes the release of cytotoxic mediators (TNF-α and IL-6) along with the activation of the pro-apoptotic protein kinase JNK and NF-κB, which are cell-injury mediators [48]. A previous study in which compounds of *C. longa* and *C. zeylanicum* were shown to be involved in the management and cure of liver injury [41,49] supports our finding. The scoring system confirms the correctness of docking by examining the lowest possible energy orientations. The docking results were justified by the in vivo results, as these receptors were dysregulated and caused oxidative stress in the case of injury, while oxidative stress was abated in the treatment groups.

Liver injury in later stages turns into liver cancer. So, it would be great if a hepatoprotective drug has the potential to treat cancer. *C. longa* and *C. zeylanicum* pure compounds have been tested against a variety of cell lines, including human cancer cell lines [50,51]. Cells were treated for 72 h, and the decreased viability of Huh7 and HepG2 cell lines and the lack of effect on the normal cell line Vero by quercetin and curcumin proved their anticancer activity, as mentioned above in Figure 3. Compounds showed significant results individually as well as in combination; a finding that makes them very useful for future use. Although both candidate drugs had the same possible target, they did not enhance the effect of each other.

Increased levels of the serum biomarkers ALT, ALP, and AST in APAP-treated mice are expressed in liver injury, and in the case of damage, these liver enzymes are discharged into the blood, which indicate extent of the liver injury. The elevated levels of these markers in the APAP-treated group showed the extent of the hepatic cell injury, and a noteworthy decrease in the levels of these markers (AST, ALT, and ALP) in the pretreatment groups in comparison with the vehicle APAP-treated group confirmed their hepatoprotective potential. Liver injury causes a decrease in the level of TAC but pretreatment with the candidate drugs showed the recovered potential of antioxidants in the serum. The antagonistic effect of TAC and TOS proved the protective effect of quercetin and curcumin. No such significant differences were observed on the liver function tests (Figure 8). The hepatoprotective effect of the screened compounds and their combination was comparable with that of silymarin, which we used as a standard drug. Our results are compatible with previous reports [52,53] stating that *C. longa*, *C. zeylanicum* are well known for their hepatoprotective abilities, as their pure compounds quercetin and curcumin have the potential to deal with oxidative stress.

The histopathological analysis also justified our in vitro results as this showed a clear effect of the different treatments on liver tissue. The APAP-treated group showed severe hepatocyte damage, inflammation, and disrupted architecture showing hepatotoxicity caused by the overdose. Pretreatment groups of quercetin and curcumin showed milder histological changes compared with APAP, suggesting a potential hepatoprotective effect. The combination-treated group showed moderate protection in the histological analysis, suggesting an additive or synergistic effect of the combined treatment. Small changes were observed in the silymarin group, suggesting a possible protective effect against liver damage. These findings suggest that quercetin, curcumin, and silymarin may have hepatoprotective properties, with silymarin having the mildest effect. Further studies are warranted to elucidate the underlying mechanisms and to determine the optimal doses and treatment durations for potential therapeutic applications.

From our observations, we conclude that the result of these treatments for liver injury follows the same trend as anticancer treatment results, proving that these are strongly associated studies. Both drugs, although having the same targets, did not contribute synergistically.

## 4. Materials and Methods

### 4.1. Chemicals and Cell Lines

Chemicals: Ascorbic acid and trichloroacetic acids were purchased from Merck and ethanol from BDH. All the other chemicals used were purchased from Sigma-Aldrich. The chemicals were supplied by local vendor of Faisalabad, Pakistan.

Cell lines: the non-cancer cell line Vero and the cancerous cell lines HepG2 and Huh-7 were kindly provided by NIBGE and NIAB, Faisalabad, respectively.

### 4.2. Collection of the Sample Plant and Its Preparation

The desired plant bark of *C. zeylanicum* (cinnamon) and rhizomes *C. longa* (turmeric) were bought from a market in Faisalabad. Plants were identified taxonomically at UAF by Dr. Mansoor Hameed, a botanist. The plant samples were washed and shade-dried before grinding them into a fine powder. In a 250 mL of Erlenmeyer flask, 15 g of each crushed plant material was soaked in 150 mL of 80% ethanol (*v*/*v*), and this was placed in an orbital shaking incubator at a speed of 300 rpm for 48 h at 25 °C. After this, by using fine cloth, concrete particles were separated from the solvent. Then, the solvent was concentrated through a rotary evaporator at 50–60 °C, and the extract obtained was stored at −20 °C [54]. 

### 4.3. In Vitro Study

#### 4.3.1. Total Phenolic and Flavonoid Contents

The total content of phenolics and flavonoids was assessed following the protocol from our published paper, Ali et al., 2021 [55].

#### 4.3.2. Antioxidative Activity

The antioxidative activity of the plant extracts was determined using DPPH and the reducing power assay by following the protocol published by Ali et al., 2021 [55] and Luqman et al., 2012 [56], respectively. The following formula was used to calculate the % DPPH radical-scavenging activity.
(1)$$\%\mathrm{DPPH}\;\mathrm{radical}\;\mathrm{scavenging}=\frac{A_{o}-A_{s}}{A_{o}}\times 100$$
where A_o_ = absorbance of the blank, and A_s_ = absorbance of the sample.

### 4.4. Cell Viability Assay

The MTT assay was performed to determine the cell viability of the cancer and non-cancer cell lines. Cell viability was assessed by the dissolution of formazan crystals. Five thousand cells were grown in each well of an ELISA plate and treated with varying concentrations (36, 7, 1.8 μg/mL of quercetin, curcumin, and doxorubicin). Then, the plate was incubated in a CO_2_ incubator for 72 h. The medium was removed, and the cells were washed with PBS, treated with 25 µL of MTT reagent, and incubated at 37 °C for 4 h. The reaction mixture was treated with 125 µL of DMSO to dissolve the formazan crystals. The absorbance was read at 570 nm using a microplate reader [57].

### 4.5. In Silico Analysis

#### 4.5.1. Screening

Molecular operating environment (MOE) analysis was used to screen for the most active compounds from the library of 40 phytochemicals present in both plants (20 from each). The 3D structures of liver injury-related receptors and compounds were retrieved from the RCSB protein databank (PDB) and PubChem, respectively. To perform the screening analysis, we mainly followed the protocol published by Mahrosh et al. [58]. The ADME analysis was carried out on 20 screened compounds for the determination of their pharmacokinetics, medicinal chemistry, and drug-likeness characteristics with Lipinski’s rule of five [59], using the open-source server SwissADME [60].

#### 4.5.2. Protein and Ligand Preparation

We selected fifteen receptors involved in liver injury from the literature, and their structure was retrieved from the RCSB protein databank (https://www.rcsb.org/structure, accessed on 25 January 2022) in PDB format. All the water and ligand molecules were removed from the protein using Discovery Studio. To prepare the proteins, the nonpolar hydrogen and Kollman charges were retained at their default settings. The selected compounds from the screening were further analyzed using the AutoDock Vina algorithm. The SDF format was converted to a PDBQT format using PyMOL 2.5.4 and AutoDock-vina 1.1.2 software. 

#### 4.5.3. Docking Analysis

AutoDock Vina was used to analyze the docking of selected compounds with the liver-injury receptors. With the help of AutoDock and PyMOL, receptors and proteins were transformed into a PDBQT format. AutoDock defines the steps to be taken to transform macromolecules. Docking was run by system command prompts [61]. 

### 4.6. In Vivo Studies 

#### 4.6.1. Mouse Model

The use of albino mice (male) aged six to seven weeks old was approved by the institutional ethical review committee of Akhuwat (FIRST Ref: AKT.FST/Misc/2022-31). After a seven-day adaptation period, the mice were split into six groups, as shown in Table 6. All except the control group received an APAP dose (200 mg/kg) by intraperitoneal injection (i.p) to induce acute liver injury [62]. After APAP treatment, the mice were starved for 24 h and then sacrificed. Blood samples were collected from the carotid arteries for analysis, and the liver was stored in formalin for histopathology [63].

#### 4.6.2. Levels of ALT, AST, ALP, and Bilirubin

Blood samples were centrifuged at 1700× *g* for 20 min. The serum was obtained as a result of centrifugation. Using the kits ALT Erba (BLT-00052), AST Human (ETI-1210021), ALK.PHOS Human (ETI-10700501), and Bilirubin Erba (BLT-0011), the levels of ALT, ALP, AST, and bilirubin were analyzed.

#### 4.6.3. Total Antioxidative Activity (TAC)

Total antioxidative capacity was determined using serum. A 15 µL aliquot of serum was mixed with 600 µL of reagent 1. Then, 60 µL of reagent 2 was mixed and incubated with the reaction mixture for 5 min at room temperature. The absorbance was read at the 630 nm wavelength.

#### 4.6.4. Total Oxidative Stress (TOS)

Total oxidative stress was determined by mixing 70 µL of serum with 450 µL of reagent 1. The first absorbance was perused while subsequently blending the mixture. Then, 22 µL of reagent 2 was blended with the reaction mixture. The absorbance was read using a spectrophotometer at the 545 nm wavelength.

#### 4.6.5. Histopathological Analysis

Hepatic tissue samples were fixed in formalin for 24 h, dehydrated with ascending grades of alcohol, and embedded in paraffin wax. Paraffin blocks were cut into 5-micron-thick slices using a microtome and stained with hematoxylin and eosin (H&E). Stained sections were viewed with a light microscope and photographed with a digital camera [67].

### 4.7. Statistical Analysis

Results are expressed as the mean ± SEM. Analysis of variance (ANOVA) and *t*-tests were performed for comparative analysis of the different groups. The level of significance was considered in terms of the *p*-value: * *p* < 0.05, ** *p* < 0.01, *** *p* < 0.001, and **** *p* < 0.0001 was considered statistically significant [68]. Graph generation and data analysis were performed using GraphPad Prism (7.01).

## 5. Conclusions

This research establishes the latest scientific foundation for determining the efficacy of multitarget compounds together with exploring more therapeutic targets for liver injury. This study uncovered hepatoprotective multitargeting compounds from the incorporation of a molecular docking approach with pharmacokinetics. Phenolics were found to be high in *C. zeylanicum* (117.5 ± 0.39 mg of Gallic acid E /g of extract) and a high content of flavonoids was found in *C. longa* (98.37 ± 0.27 mg QE/g of Extract). Furthermore, our findings propose quercetin and curcumin as promising and viable therapeutic drugs to reduce the incidence of liver injury. It has been concluded that the plants *C. longa and C. zeylanicum* have anticancer and hepatoprotective potential due to the presence of their bioactive compounds, quercetin and curcumin. These compounds showed effective results in combination as well as individually. Future studies will be able to report on the mechanism of action of combined bioactive compounds screened through in silico analysis for the treatment of various diseases.

## Figures and Tables

**Figure 1 molecules-29-00428-f001:**
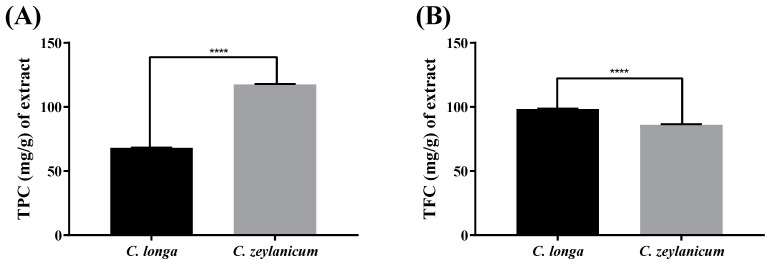
Total phenolic and flavonoid content. (**A**) TPC was determined in *C. longa* and *C. zeylanicum*. (**B**) The content of flavonoids was determined in *C. zeylanicum* and *C. longa*. Both are expressed in unit milligram of Gallic acid equivalents per gram of each extract. The level of significance is represented by the *p*-value (**** = *p* < 0.0001), and the *t*-test was performed to determine this value.

**Figure 2 molecules-29-00428-f002:**
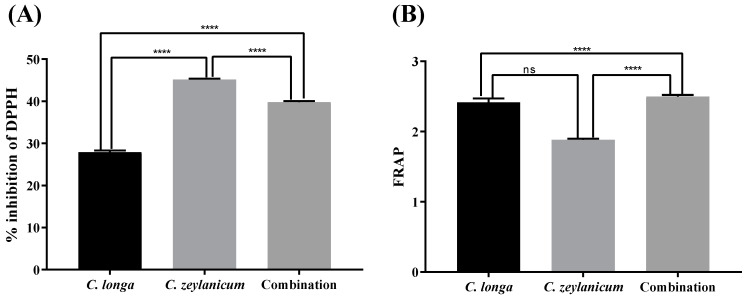
Antioxidative potential of plant extracts and their combination. (**A**) The % inhibition of DPPH by combined and individual extracts *C. longa* and *C. zeylanicum* was measured at 515 nm. (**B**) Reducing power of plant extracts from *C. longa* and *C. zeylanicum*. The significance level was determined by applying analysis of variance (one-way ANOVA) and Tukey’s multiple comparison tests and is represented by the *p*-value (**** = *p* <0.0001). ns showed Non-significant value.

**Figure 3 molecules-29-00428-f003:**
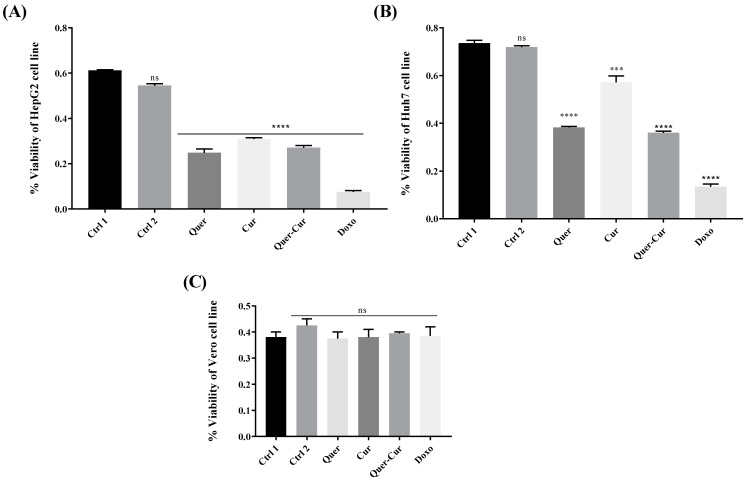
Effect of the pure compounds quercetin and curcumin along with their combined extract on the viability of cells. (**A**) Cancerous cell line HepG2; (**B**) Cancerous cell line Huh-7; (**C**) Non-cancer cell line Vero. Ctrl 1 represents the control with distilled water; in Ctrl 2, 0.1–0.2% DMSO was present. The level of significance is represented by the *p*-value *** *p* < 0.001, and **** *p* < 0.0001. ns represents non-significance. The significance level was determined by applying analysis of variance (one-way ANOVA) and Tukey’s multiple comparison tests. Quer: quercetin; Cur: curcumin; Doxo: doxorubicin.

**Figure 4 molecules-29-00428-f004:**
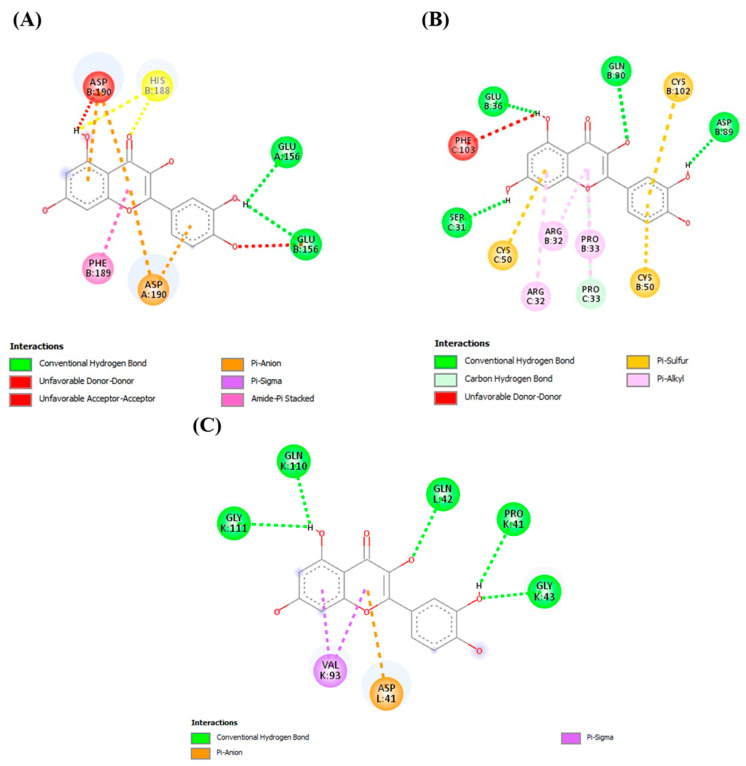
Docking of quercetin with the liver-injury-related receptors/enzyme CYP2E1, MAPK, and TLR4. (**A**) Quercetin interaction with CYP2E1. (**B**) Quercetin interaction with MAPK. (**C**) Quercetin interaction with TLR4.

**Figure 5 molecules-29-00428-f005:**
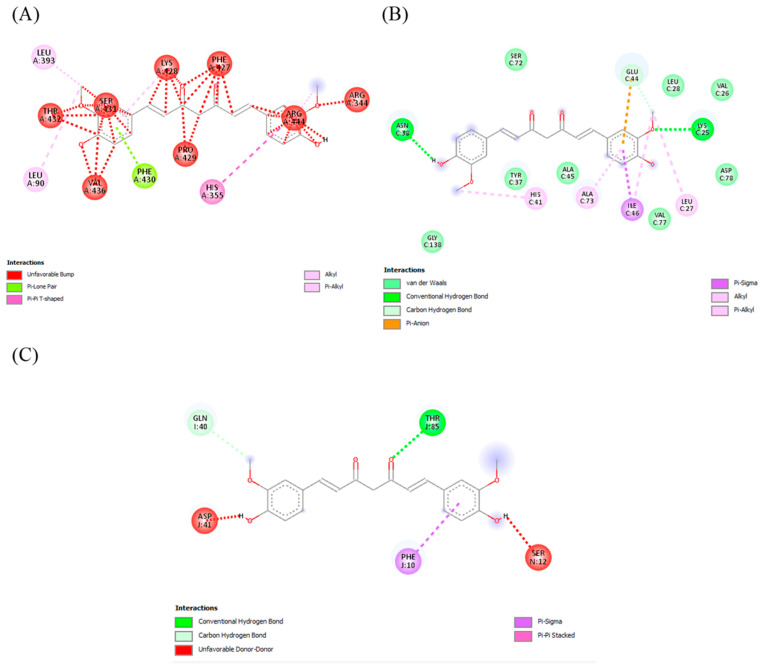
Docking of curcumin with the liver-injury-related receptors/enzyme CYP2E1, MAPK, and TLR4. (**A**) Curcumin interaction with CYP2E1. (**B**) Curcumin-binding pattern with MAPK. (**C**) Curcumin interaction withTLR4.

**Figure 6 molecules-29-00428-f006:**
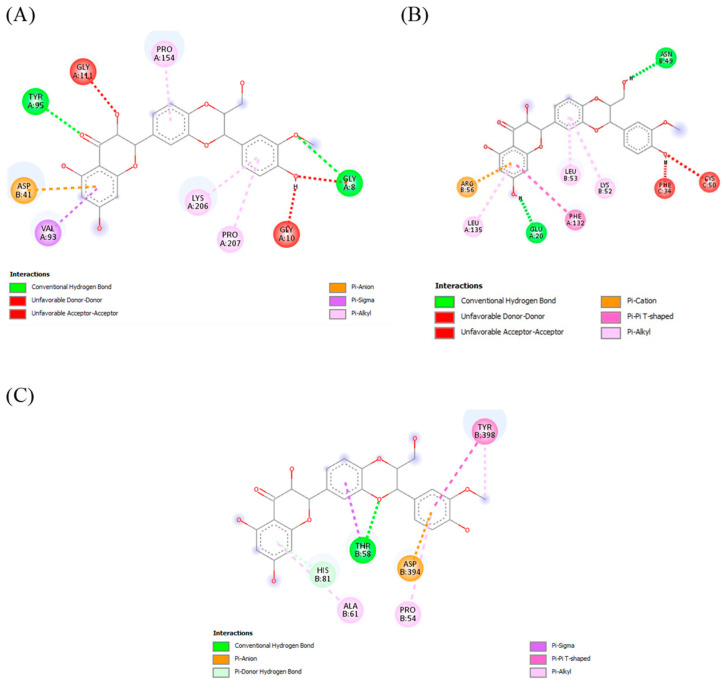
Docking of silymarin with the liver-injury-related receptors/enzyme CYP2E1, MAPK, and TLR4. (**A**) Silymarin interaction with CYP2E1. (**B**) Silymarin interaction with MAPK. (**C**) Silymarin interaction with TLR4.

**Figure 7 molecules-29-00428-f007:**
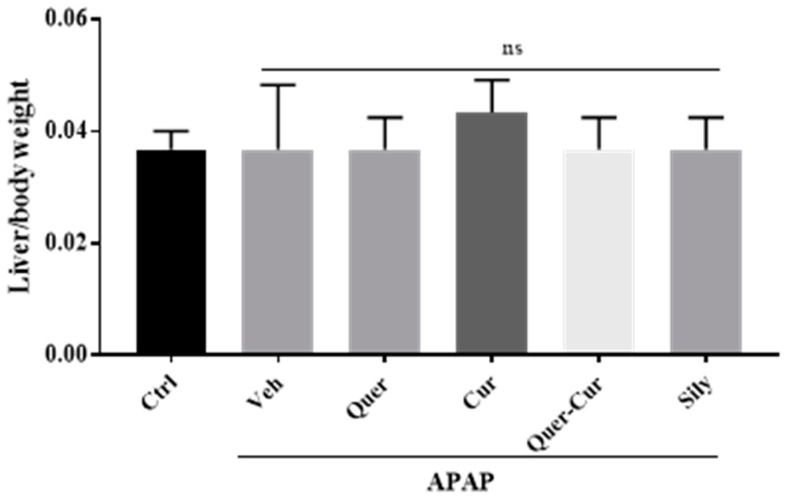
Liver to body weight. Comparative analysis of the liver to body weight ratio showed non-significant changes when compared with the control. Veh: vehicle (APAP); Quer: quercetin; Cur: curcumin; Sily: silymarin. ns showed level of significance (non-significant).

**Figure 8 molecules-29-00428-f008:**
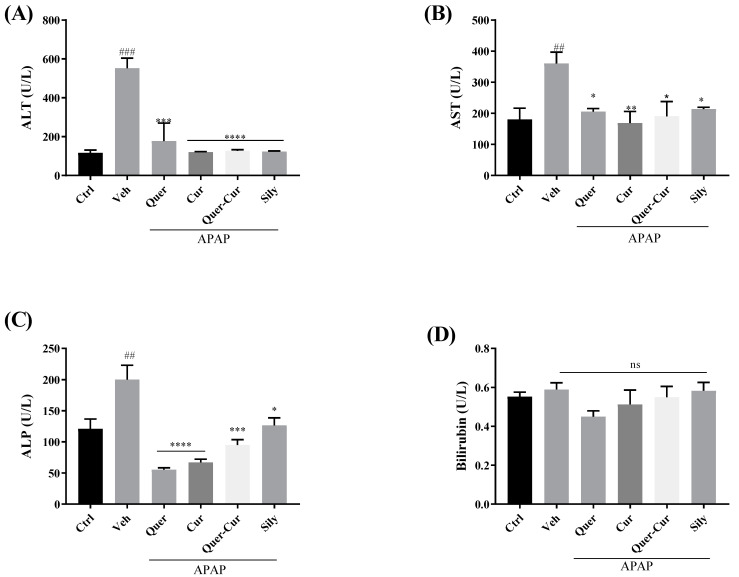
Effects of pure compounds quercetin, curcumin, their combination, and the standard drug silymarin on the serum biomarkers (**A**) aspartate aminotransferase (AST), (**B**) alanine aminotransferase (ALT), (**C**) alkaline phosphatase (ALP), and (**D**) bilirubin. The level of significance is represented by the *p*-value: * *p* < 0.05, ** *p* < 0.01, *** *p* < 0.001, and **** *p* < 0.0001; ## *p* < 0.01, ### *p* < 0.001 showed vehicle comparison to negative control. The significance level was determined by applying analysis of variance (one-way ANOVA) and Tukey’s multiple comparison tests. Veh: vehicle (APAP); Quer: quercetin; Cur: curcumin; Sily: silymarin.

**Figure 9 molecules-29-00428-f009:**
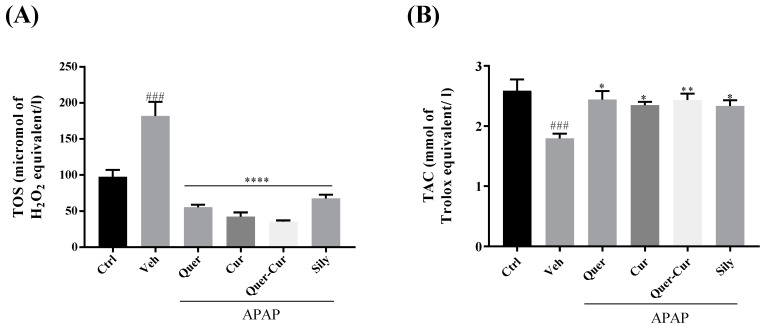
Effects of the pure compounds quercetin, curcumin, their combination, and the standard drug silymarin on oxidative stress markers in tissue samples. (**A**) Total oxidative stress. (**B**) Total antioxidant capacity. The level of significance is represented by the *p*-value: * *p* < 0.05, ** *p* < 0.01 and **** *p* < 0.0001. ### *p* < 0.001 showed vehicle comparison to negative control The significance level was determined by applying analysis of variance (one-way ANOVA) and Tukey’s multiple comparison tests. Veh: vehicle (APAP); Quer: quercetin; Cur: curcumin; Sily: silymarin.

**Figure 10 molecules-29-00428-f010:**
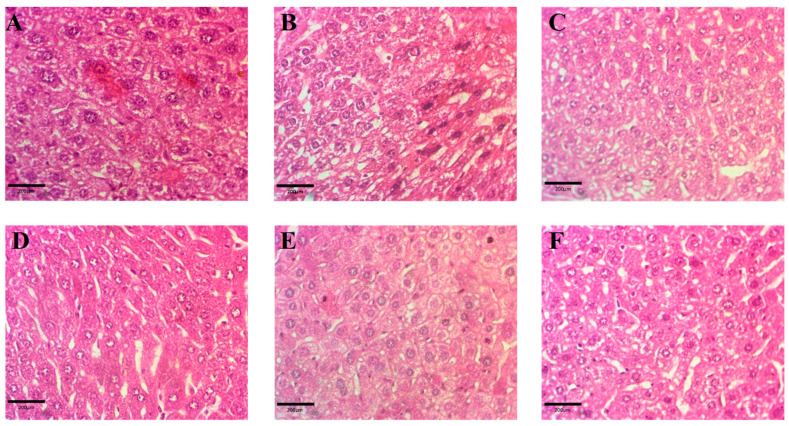
Digital images of liver tissues showing the effect of different treatments. (**A**) Control group; (**B**) APAP-treated group; (**C**) Quercetin-APAP treated group; (**D**) Curcumin-APAP treated group, (**E**) Quer. + cur-APAP treated group; (**F**) Silymarin-APAP treated group. Scale bar represents 200 µm.

**Table 1 molecules-29-00428-t001:** Percentage yield from *Curcuma longa* and *Cinnamon zeylanicum*.

Sample Plant	Part of Plant	Solvent	% Yield ± SEM
*C. longa*	Rhizomes	80% Ethanol	12.7 ± 0.09
*C. zeylanicum*	Bark	80% Ethanol	10.84 ± 0.09

**Table 2 molecules-29-00428-t002:** Total content of flavonoids and phenolics in *Curcuma longa* and *Cinnamon zeylanicum*.

Sample	TPC (mg of Gallic Acid E/g of Extract)	TFC (mg QE/g of Extract)
*C. longa*	68.0 ± 0.17	98.37 ± 0.27
*C. zeylanicum*	117.5 ± 0.39	86.11 ± 0.47

**Table 3 molecules-29-00428-t003:** Percentage yield of *Curcuma longa* and *Cinnamon zeylanicum*.

Sample Name	DPPH %	Reducing Power %
*C. longa*	27.89 ± 1.33	2.41 ± 0.17
*C. zeylanicum*	45.16 ± 0.66	1.88 ± 0.05
Combination	39.81 ± 0.73	2.49 ± 0.07

**Table 4 molecules-29-00428-t004:** Important pharmacokinetic factors for the compound bioavailability and drug-like characteristics of the chosen compounds.

Compound	MW	HBD	HBA	Nrotb	Log*P*	A	Violations
Curcumin	368.38	2	6	8	3.37	102.8	0
Quercetin	302.24	5	7	1	1.99	78.04	0
Calebin A	384.38	2	7	8	2.88	103.89	0
Rutin	610.52	10	16	6	−1.51	141.38	3

HBD: hydrogen-bond donors; HBA: hydrogen-bond acceptors; Nrotb: number of rotatable bonds; MW: molecular weight; HBD: number of hydrogen-bond donors; A: molar refractivity; Log*P*: logarithm of the partition coefficient (octanol/water).

**Table 5 molecules-29-00428-t005:** Property profiles of specific compounds concerning liver-injury-related proteins.

Ligand	Receptor/Enzyme	S-Score	Interacting Amino Acids
Quercetin	CYP2E1	−7.4	A chain (Asp 190, Glu 156), B chain (His 188, Asp B 190, Phe 189)
	MAPK	−9.2	B chain (Glu 36, Gln 90, Cys 102, Asp 89, Cys 50, Pro 33, Arg 32) C chain (Phe 103, Ser 31, Cys 50, Arg32, Pro 33)
	TLR4	−8.2	K chain (Gly 111, Gln 110, Pro 41, Gly 43, Val 93), L chain (Asp 41, Gln 42)
Curcumin	CYP2E1	−6.1	A chain (Leu393, Thr 432, Ser 431, Lys 428, Arg 344, Arg 444, His 355, Phe 427, Leu 90, Val 436, Phe 430, Pro 429)
	MAPK	−5.5	C chain (Asn 38, Glu 44, Lys 25, Leu 27, Ile 46, Ala 73, His 41)
	TLR4	−7.3	Gln I:40, Ser N:12, J chain (Asp 41, Phe 10, Thr 85)
Silymarin	CYP2E1	−8.9	A chain (Gly 8, Gly 10, Pro 207, Lys 206, Val 93, Tyr 95, Gly 111, Pro 154), Asp B:41
	MAPK	−9.5	A chain (Leu 135, Glu 20, Phe 132), B chain (Arg 56, Leu 53, Lys 52, Asn 49), C chain (Phe 34, Cys 50)
	TLR4	−9.0	B chain (Tyr 398, Asp 394, Pro 54, Thr 58, Ala 61, His 81)

**Table 6 molecules-29-00428-t006:** Animal groupings for the in vivo studies.

Group	Group Description	Treatment	Dose Conc./Duration Time	References
Group 1	Control group	Normal saline	For three weeks	[64]
Group 2	Induced toxicity group	Received APAP	APAP 200 mg/kg body weight	[62]
Group 3	Protective group (Quercetin)	APAP + QUE	QUE (20) mg/kg for 21 days	[65]
Group 4	Protective group (Curcumin)	APAP + CUR	CUR (50) mg/kg for 21 days	[66]
Group 5	Protective group (Combination)	APAP + QUE + CUR	MIX (1:1) mg/kg for 21 days	--
Group 6	Protective group (Silymarin)	APAP + SILY	SILY (50) mg/kg for 21 days	[66]

All groups except the control group received APAP (200 mg/kg body weight) to induce hepatotoxicity after pretreatment.

## Data Availability

The data presented in this study will be available on request from the corresponding author.

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
