# Peer review of "Screening of Multitarget Compounds against Acetaminophen Hepatic Toxicity Using In Silico, In Vitro, and In Vivo Approaches"

_molecules, 2024, doi:10.3390/molecules29020428_

Round 1

Reviewer 1 Report

Comments and Suggestions for Authors

The paper is well written, and the objectives are clear. However, the paper needs some editing. All the plant names should be written in italics. some values must be included in the abstract. There is no need to present the same result in a Figure and in the Table. The positive control must be included in all assays. The authors must consult on how to write the plant name in teh sentence e.g. C. zeylanicum .

L17: APAP must be written in full

All the plant names must be written in italics and if not at the beginning of a sentence

There is duplication of results presented as a table and also as a figure

Include the positive controls for all the assays

Indicate the sample size for all the assays used

L383: Check how CO2 is written

There should be a space between the unit and the number correct this throughout.

Comments on the Quality of English Language

Need some editing

Author Response

L17: APAP must be written in full

Acetaminophen has been written

All the plant names must be written in italics and if not at the beginning of a sentence

italicized 

There is duplication of results presented as a table and also as a figure

Include the positive controls for all the assays

in anticancer study doxorubicin as positive control and in invivo experiments silymarin acting as a positive control

L383: Check how CO2 is written

corrected CO2

There should be a space between the unit and the number correct this throughout.

corrected

Reviewer 2 Report

Comments and Suggestions for Authors

-In the introduction section, a wider range of bioactive properties of the curcuma longa compound should be mentioned. These properties of curcumin, which is very popular with its antioxidant, antibacterial and anticarcinogenic properties, should also be emphasized and especially the results obtained should be compared. The following article will be helpful.

Biocidal Activity of Bone Cements Containing Curcumin and Pegylated Quaternary Polyethylenimine

T Eren, G Baysal, F DoÄŸan

Journal of Polymers and the Environment 28, 2469-2480

-In the introduction section, the superior features of this research compared to the literature need to be explained in more detail.

-. In 250 ml of Erlenener 362..... please correct "ml" to "mL" throughout the manuscript

-d with 25 µl of MTT reagent, and incubated at 37°C for 4 h. 384

The reaction mixture was treated with 125 µl of DM.... similar problem??????

-In the conclusion section, briefly mention the bioactive performances (numerical) of the components according to the analysis results.

-Please explain for what period the cell viability analyzes were performed, because this period is very important for interpreting the results. (24h , 48h or 72h which one?????

-There is a need for a table in the discussion section comparing the analysis findings with the literature findings

Comments on the Quality of English Language

Minor editing of English language required

Author Response

-In the introduction section, a wider range of bioactive properties of the curcuma longa compound should be mentioned.

the properties of curcumin and quercetin has been mentioned in introduction section

-. In 250 ml of Erlenener 362..... please correct "ml" to "mL" throughout the manuscript

corrected

-d with 25 µl of MTT reagent, and incubated at 37°C for 4 h. 384

corrected

The reaction mixture was treated with 125 µl of DM.... similar problem??????

corrected

-In the conclusion section, briefly mention the bioactive performances (numerical) of the components according to the analysis results.

mentioned accordingly

-Please explain for what period the cell viability analyzes were performed, because this period is very important for interpreting the results. (24h , 48h or 72h which one?????

the cells were treated and incubated for 72 h . we have mentioned this in methodology and discussed 

Reviewer 3 Report

Comments and Suggestions for Authors

Reviewed publication by Ali M et al. "Screening of Multitarget Compounds Against Acetaminophen Hepatic Toxicity Using In-Silico, In-Vitro, and In-Vivo Approaches" shows the influence of two compounds on the proper functioning of the liver. However, the work has several errors and shortcomings, listed below.

Major.

1. CYP2E1 or different CYP enzymes are not a receptor. Please correct it in the manuscript (text and tables).  

2. Additionally, there are no deviations in the figures. There is also no information about the deviation under each graph. Statistical significance in graphs (p-value) should be more visible. In the description of the all figures does not elaborate on the appropriate abbreviations used. There is also no description of the statistical test used. Please complete it. 

3. There is an error in the numbering of tables. Please correct. 

4. Figure 3. In the description of the method (4.3), four concentrations of the tested compounds were used, and the figures show only 2 concentrations, which do not correspond to those in the method. Please supplement and explain.

Minor:

1. No explanation of abbreviations in Figures (e.g. Quer, Sily, Veh). The text also lacks explanations of certain abbreviations (e.g. line 327).

2. Figure 9. Lacks  explanation of p-value for **

3. Table 1. Why different parts of plants were used? Please of explanation. 

4. Data in the figures are presented as Mean ± SEM. I would strongly recommend presenting as „Mean, SD“. I also lack justification for using the Student t-test for statistical evaluation (data normally distributed? Use of nonparametric Wilcoxon sign test?) 

5. Line 148. Misspelling in HepG2 cell line 

6. Line 287-288. Cytochrome P450 is correct , and CYP1A2, CYP2C9, CYP2D6, CYP2C19 and CYP3A4 enzymes name. 

7. Table 7. Why were concentrations of tested substances and drugs selected for in vivo tests? Lacks explanation and references.

Author Response

CYP2E1 or different CYP enzymes are not a receptor. Please correct it in the manuscript (text and tables).  

corrected

2. Additionally, there are no deviations in the figures. There is also no information about the deviation under each graph. Statistical significance in graphs (p-value) should be more visible. In the description of the all figures does not elaborate on the appropriate abbreviations used. There is also no description of the statistical test used. Please complete it. 

corrected

3. There is an error in the numbering of tables. Please correct. 

corrected

4. Figure 3. In the description of the method (4.3), four concentrations of the tested compounds were used, and the figures show only 2 concentrations, which do not correspond to those in the method. Please supplement and explain.

the concentrations mentioned in 4.3 are of different compounds. Quercetin, curcumin and doxorubicin. there was minor mistake which has been removed.

Minor:

  1. No explanation of abbreviations in Figures (e.g. Quer, Sily, Veh). The text also lacks explanations of certain abbreviations (e.g. line 327).

abbreviations added

2. Figure 9. Lacks  explanation of p-value for **

explained 

3. Table 1. Why different parts of plants were used? Please of explanation. 

before selecting the part of plant we have reviewed literature. and these parts have been reported to have high content of TPC and TFC value. and have high biological activities 

4. Data in the figures are presented as Mean ± SEM. I would strongly recommend presenting as „Mean, SD“. I also lack justification for using the Student t-test for statistical evaluation (data normally distributed? Use of nonparametric Wilcoxon sign test?) 

we understand your suggestion to consider alternative nonparametric tests such as the Wilcoxon sign test. While appreciating the importance of nonparametric methods, our data, after rigorous evaluation, adheres to the assumptions necessary for parametric tests. Employing the t-test allows for a more sensitive analysis, considering the nature of our dataset and ensuring statistical power to detect meaningful differences between groups.We believe that retaining the "Mean ± SEM" presentation and utilizing the Student t-test aligns with the nature of our data and the standards in the field, allowing for a comprehensive and accurate representation of our findings.

5. Line 148. Misspelling in HepG2 cell line 

corrected

6. Line 287-288. Cytochrome P450 is correct , and CYP1A2, CYP2C9, CYP2D6, CYP2C19 and CYP3A4 enzymes name. 

corrected

7. Table 7. Why were concentrations of tested substances and drugs selected for in vivo tests? Lacks explanation and references.

we have reviewed literature for best concentrations by keeping lethal dose of these compounds in mind.. references were missed due to technical error ....now added

Round 2

Reviewer 3 Report

Comments and Suggestions for Authors

The authors made most of the revisions to the manuscript. However, there are still some errors that need to be corrected.

1. The figures have not been further improved. are too small and statistical significance is almost invisible. Please correct.

2. In Tables S2, S5 and S6, there is still information that CYP enzymes are receptors.

3. Lines 319, 320 Cytochrome names have not been corrected

Author Response

  1. The figures have not been further improved. are too small and statistical significance is almost invisible. Please correct.

figure quality have been updated

2. In Tables S2, S5 and S6, there is still information that CYP enzymes are receptors.

corrected

3. Lines 319, 320 Cytochrome names have not been corrected

corrected